# Challenges for Social Participation in Conservation in the Biocultural Landscape Area in the Western Sierra of Jalisco

Oscar Alberto Maldonado Ibarra, Rosa María Chávez-Dagostino *, Myrna Leticia Bravo-Olivas and Rosío T. Amparán-Salido

Centro Universitario de la Costa, Av. Universidad de Guadalajara, Puerto Vallarta 48280, Jalisco, Mexico; oscar.maldonado@academicos.udg.mx (O.A.M.I.); myrna.bravo@academicos.udg.mx (M.L.B.-O.); rosio.amparan@cuc.udg.mx (R.T.A.-S.)
* Correspondence: rosa.cdagostino@academicos.udg.mx

**Abstract:** The protection of biocultural heritage has generated alternative proposals for the conservation of rural areas. Varied organizations collaborate in a pioneering conservation model, the Biocultural Landscape (BL), where local participation is paramount, that operates in the Western Sierra of Jalisco. The objective of this work was to analyze social participation, conditions, and characteristics of the model based on the WWF and IUCN guidelines. Information about the context, management, and planning was collected and synthesized. The data of territorial management, conservation, knowledge, and local conflicts about participatory processes were collected from 12 stakeholders and analyzed with ATLAS.ti software. It was found that, although local people are familiar with the concept of the protected natural area and the BL model, they cannot clearly identify its objective. The most informed are the interested population that collaborates closely. There are conflicts of interest between those who collaborate directly with the BL and those who do not, which have been resolved through the active participation of the different levels of government and experts who have intervened as mediators. Environmental awareness about the importance of conservation has been achieved by integrating the communities. Given that it is not a restrictive protection model and the rules were created in conjunction with the community, local participation is encouraged.

**Keywords:** biocultural; social participation; natural protected area; conservation

## 1. Introduction

Despite advances in conservation, multiple challenges are being faced around the world to achieve it, and alternatives are being sought where society and government join efforts to accelerate and ensure the process that leads to sustainable use of natural resources. However, in a world dominated by a western view, where humans are not part of nature but above it, conservation achievements are limited even in protected natural areas. Some have concluded that, to date, protected areas around the world often fail in the aim of achieving ecosystem conservation and of improving local people's wellbeing [1]. This has made necessary other forms of conservation and management of nature [2], which could be considered innovative. Understanding the characteristics of the interdependent society–nature relationship is necessary to achieve conservation, which is manifested through values, symbols, and worldviews [3]. The organization, functioning, vitality, and resilience of ecosystems and human communities are mutually linked, a fact that has been recognized since the 1990s by various authors [4], which implies the diversity of life in all its manifestations, biological as well as cultural and linguistic, related in a complex way in an adaptive socio-ecological system.

Although there is debate about the biocultural concept, Lindholm and Ekblom [5] established that the understanding of cultural landscapes is the result of long-term biological and social relationships, which shape the biological and material characteristics of

the landscape as well as memory, experience, and knowledge. Biocultural diversity denotes the diversity of life in its multiple manifestations: biological, cultural, and linguistic, framed in sociocultural systems [6]. The interdependence between biological and cultural diversity has been developed over time through adaptive processes [7], and the rapid loss of both around the world has led to a concern about its effects on the achievement of sustainability objectives [8]. Social participation is, then, required to achieve conservation, and designing natural protected areas is one of the common strategies for biological and cultural conservation. However, there is evidence that the limited participation of local populations in conservation proposals generates conflict. On the other hand, participation increases the credibility of government authorities, reduces potential conflicts between the parties, improves the quality and quantity of information that flows better in the system, reduces the power of some dominant actors, and improves the decision-making process [9]. Given the limited achievements of participation in environmental governance, rather than focusing on improving the participatory process, some authors tried to understand its failures and concluded that the institutional and political context is a determining factor and includes the provisions for participation in the directives, lack of policy integration, and lack of political assimilation of the results of participation [10].

Generally, the topic of social participation in works on natural protected areas is scarce and addresses issues about the influence of local governments, the limited participation of communities, and conservation as a conflictive issue [11–13].

Even when the concept "biocultural" is not used as associated with landscape and related to natural protected areas, it is always implicit. There are examples where natural and cultural heritage are considered with the same importance. In Slovakia, the agricultural landscape was studied related to the wine-growing landscape, herbaceous-pastoral landscape, and livestock landscape, and concluded that the participation of the local population has been fundamental as a source of information and management in the concern for the extinction of agricultural landscapes [14]. In the Netherlands, measures related to biocultural have been relevant for the management of natural parks and the increase of local participation [15] in areas such as nature conservation, including individual and collective activities related to studies, training of guides, and volunteers and recreative activities related to nature.

Agriculture has been relevant to the biocultural issue due to its centuries-old practices that are now focused on sustainable activities and in some countries such as Morocco, these sites have been considered World Heritage Sites by UNESCO and part of the MAB program [15,16].

As in many places in the world, natural protected areas in Mexico face various problems, such as uncoordinated public policies, local conflicts due to the use and control of natural resources, exclusion of community participation, and damage to populations for obtaining their economic livelihood [17–20]. These complications result in the need to generate strategies according to the characteristics of the territory, in favor of local integration and the protection of the biocultural heritage. These are crucial for the proper functioning of the territory and appeal to a complex socio-ecological vision that entails learning and collaboration with a large number of actors in a region [21–23].

The French and Mexican governments explored the possibility of adapting local governance schemes that would support biological and cultural diversity conservation work in Mexico promoting sustainable rural development. This was done through the adaptation of the French Regional Natural Park model [24] that would give rise to the figure of BP in Mexico as an innovative model of territorial valorization [25].

In the western Sierra of Jalisco in Mexico this regional model was called BL, whose challenge is to achieve conservation in a broad sense through sustainable development and where some questions arise: Does this model surpass the traditional model of protected natural areas implemented by governments? What challenges does it face related to social participation? In this context, the model and its characteristics in the State of Jalisco in Mexico are described below.

*1.1. Biocultural Landscape Model and Local Participation*

The interest in the rural world, in the face of the failure of top-down development models, led to the emergence in the 1990s of the community initiative known as LEADER, based on a territorial approach, the creation of new participatory local government structures, and decentralized management [26].

The European Landscape Convention was the first international convention focused on this issue, derived from the concern to achieve sustainable development based on a balanced relationship between social needs, the economy, and the environment, thus, a central element of the quality of life of human populations [27]. In this context, a predecessor of the BL can be considered the IUCN category V Landscape, unique among the different categories due to its emphasis on people–nature interaction, which recognized the need to design areas which were different from those subject to strict protection, but were also economically, socially, culturally, and environmentally important. This category considers support for human communities and the sustainable use of natural resources, which has been important for developing countries facing problems of poverty and protection of local culture and nature, especially in rural communities [28].

Given the various conflicts associated with protected natural areas and the participation of local populations, the BL is a territory recognized for its biocultural and landscape value, in which all stakeholders (government and inhabitants) collaborate synergistically around a concerted sustainable development project through which, as result of protection and valorization of the biocultural heritage, economic development is promoted [29]. Territorial management is defined and adopted by municipal governments, the state, federal governments, and the representative bodies of social groups involved. Its main virtue is the active participation of local actors, mainly producers and traders, as well as support for the conservation of natural areas by the communities' own decisions and not due to a decree made by outside interests. Sustainable resource management is sought in the area to strengthen governance mechanisms, financing processes, and productive chains and to support the institutionalization of governance in the territory through local production [30]. The region provides environmental services to local communities as well as food, commercial products, climate, cultural, and tourism services.

The BL model is a Mexican proposal of governance [24] that bases its operation on a Territorial Management Agreement which generally reflects the development aspirations of a region, aspirations which were collected in various ways and have three main orientations: (1) Environment and territory; (2) Society and culture; and (3) Economy and solidarity [31]. For the first, the aspiration is that inhabitants and various socio-productive sectors preserve biodiversity, goods, and services provided by the natural environment. Related to society and culture, the inhabitants, visitors, and diverse socio-productive sectors recognize and value their cultural heritage and territorial identity, recovering and using their traditional knowledge. The third states that production systems are sustainably developed and managed to ensure equity and justice in institutional, commercial, and environmental relations.

Each orientation has axes and involves measures, provisions, and actions that also have transverse orientations that encourage the achievement of each of the axes: Capacity building; Collaborative and differentiated management for all; Education, linkage, and research. It can be assumed that this would be the framework for stakeholder participation and interaction with the territory in question, which, unlike conventional natural protected areas, generally lacks prior agreement and a common vision.

The model has been publicized as successful and as a mechanism for sustainable territorial management, based on social participation, which is a key aspect of territorial governance, both for the sustainable use of resources and for improving living conditions. However, this conception of management has been questioned as it is considered an "emerging form of social control under the premise of participation" ([20], p. 214) that can lead to the homogenization of socio-cultural differences in the long term.

A new question arises at this point: has the model succeeded in improving participation to achieve conservation objectives in the Western Sierra of Jalisco? We hypothesized that the BL model has improved social participation, unlike traditional natural protected areas generally designed by governments which linearly and hierarchically exclude local participation.

### 1.2. Study Area

The BL area of 245,000 hectares partially encompasses four rural municipalities in the Western Sierra of Jalisco under a voluntary state conservation decree: San Sebastián del Oeste, Mascota, Talpa de Allende, and Atenguillo (Figure 1). The first three are home to "Magical Towns", which is the name of a federal program that recognizes their tourism value related to their natural and cultural heritage.

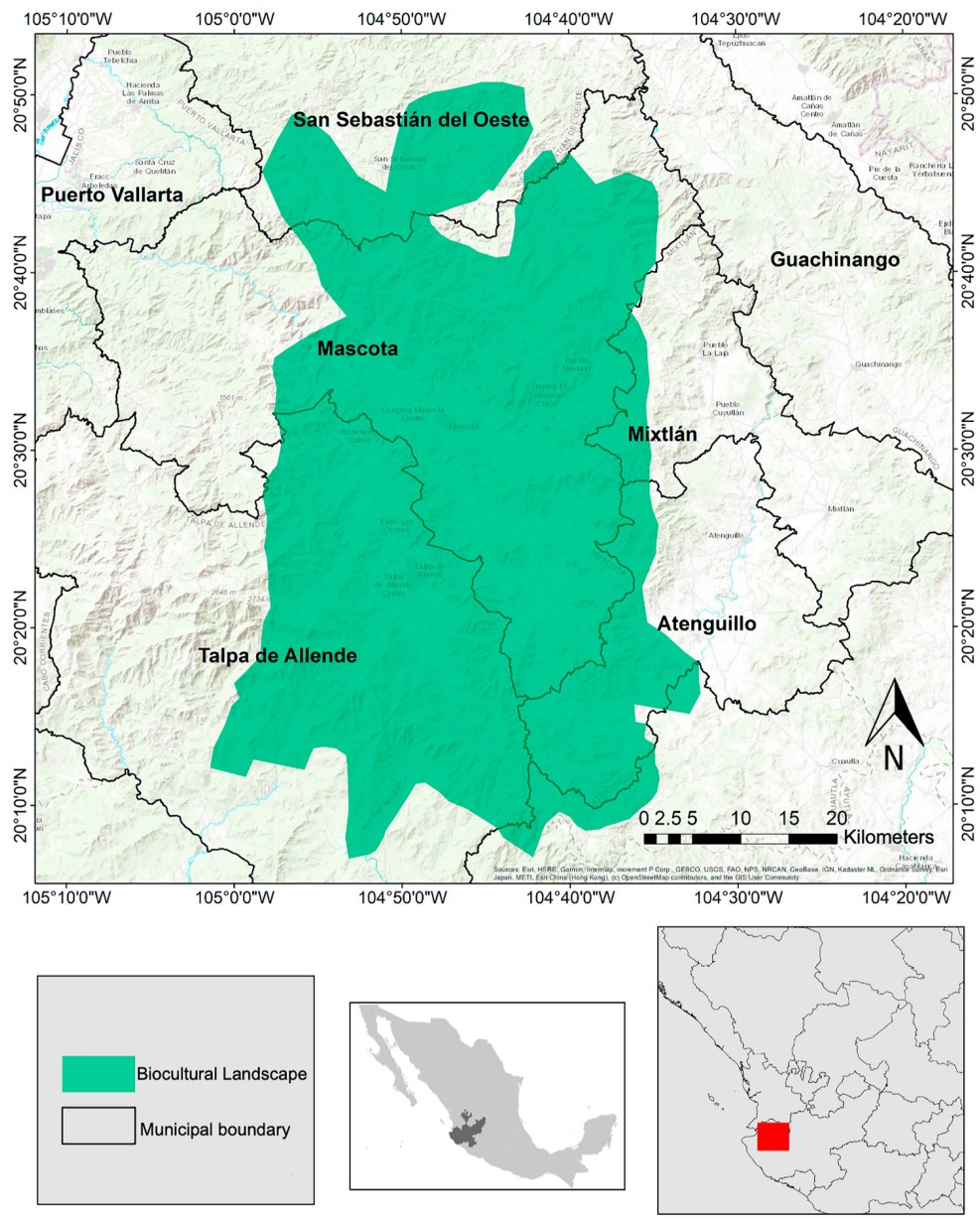

**Figure 1.** Geographical location of the Biocultural Landscape area at Western Sierra (in red) of Jalisco (in gray).

The total population in BL is 31,786, where the main economic activities are based on agriculture, cattle ranching, commerce and tourism. [32–35].

The natural heritage of this region is made up of mountains (64%) whose maximum altitude is 2760 m above sea level and which form an important hydrological basin that drains into the Pacific. The main land uses are (a) flora and fauna, dominated by maple and pine-oak forests with species such as white-tailed deer and jaguar, occupying 42.1% of the region's total area; (b) forestry (28.26%); (c) natural area (17.47%); (d) agriculture (11.21%); (e) tourism (0.71%); and (f) human settlements (0.24%) [36]. The cultural heritage is represented by archeological vestiges, religious constructions, pilgrimages, and mines [37–39].

## 2. Methods

### 2.1. Evaluation of the Study Area

The BL area was evaluated for its context, management and planning according to IUCN's World Commission for Protected Areas (WCPA) and World Wildlife Fund (WWF) tool to track the management effectiveness of protected areas [40,41]. It is a framework to guide the assessment systems and the dissemination of results based on the premise that the good management of protected areas would be the product of a process that includes the analysis of the context of existing values and threats, the revision of the planning and allocation of resources (inputs), management actions (processes) and the eventual production of services and goods (products). A questionnaire with 20 items was adapted to the BL model as a guide to interview those in charge of the BL area, who also provided documents to demonstrate the arguments.

Two sections were included; the first contains information on the characteristics and design of the BL including status, regulations, laws, objectives, management plans, and natural characteristics. One of the items in this section requires a comment on critical habitat/ecosystems. This refers to areas that present particular conditions for the survival of a species or population and, so, require special management and protection. The second section evaluated infrastructure and participation, economic aspects, facilities, human capital, education, and local participation. Their comments on each item in both sectors were synthesized and then analyzed separately in a matrix, where the staff in charge expressed their opinions.

### 2.2. Stakeholders in the BL Model

As part of the second stage, 12 key stakeholders related to the BL, whose participation was considered indispensable and mandatory to achieve its purposes and goals, were identified [42]. Stakeholders are understood here as individuals who represent different groups that can affect the achievement of the objectives of the BL as an organization or who are affected by the achievement of its objectives [43].

The steps followed for the identification of key stakeholders were:

1. Collecting of information from key stakeholders that could be directly or indirectly related to the project (field and desk work).
2. Design of a database of key stakeholders according to local, municipal, and state levels.
3. Analysis of the interaction between stakeholders.
4. Stakeholder mapping.
5. Initiation of outreach, communication, and intervention strategies.

The resulting selected stakeholders represent local governments related to tourism and the environment, culture, academic, and commerce activity who were interviewed during the period of June 2019 to March 2020. The first contact was through the staff responsible for BL management and with a chain sampling including local and regional actors related to the BL area of the four municipalities, respecting the diversity of roles in the area (Table 1).

**Table 1.** Roles of the interviewed actors.

| Actors | Role |
| --- | --- |
| 1 | Municipal government official (tourism sector) |
| 2 | BL staff |
| 3 | Local artisan |
| 4 | Municipal government official (tourism sector) |
| 5 | Local representative of state environmental institution |
| 6 | Municipal government official (culture and tourism) |
| 7 | Municipal government official (ecology) |
| 8 | Local merchant |
| 9 | BL staff |
| 10 | Municipal government official (ecology) |
| 11 | Academic |
| 12 | BL staff |

The interview included 14 open questions and was structured considering the themes of territorial management, conservation, knowledge, and local conflicts about participatory processes between the local population and natural protected areas. The results were processed with ATLAS.ti v8 software; 33 categories (codes) were created and distributed into three groups: local conflicts (6), conservation and knowledge (11) and territorial management (16). Each code represents a label assigned to a response, which contains information on the topic addressed by the code group. Subsequently, responses with common themes were described.

## 3. Results

### 3.1. Characteristics and Design of the Biocultural Landscape Area in the Western Sierra of Jalisco

Context: Evaluation of the area states that the territory designed as BL is in process of recognition by the government because the actual environmental legislation in Mexico does not include a protected area with those characteristics. The existing mechanisms and rules are established in a document called "Territorial Chart" that functions like a management program for a natural protected area and where regulations and norms are described as well as a monitoring of the goals, plans, and results of the BL. Increased operational capacity could improve the implementation of legislation. This is a legal issue and amendments to the law are expected. Although the area is not physically demarcated, the boundaries are known, but work is underway to install infrastructure that will allow local people and visitors to know where the area is located and where to go. Given that it is an area of biological importance, there is some knowledge of the ecosystems included, but not enough (Table 2).

Management: The area has a high conservation value due to its ecosystem composition. The value of the territory is a function of connectivity with other areas in the region such as Manantlán, Cacoma, and Sierra de Vallejo. The most important function is connectivity between ecosystems because it allows the passage of certain species that require a wide space to move around (Table 2).

Planning: The objectives are set out in the "Territorial Chart" and foreseen for a 15-year period; the orientations and strategies to be followed are determined, and they are socialized with the socio-productive. Its approach and characteristics are favorable, as the conservation objectives promote sustainable land management. Currently, it is being analyzed and receiving feedback before it is approved and validated. An operative annual plan is elaborated each year and the BL council approves it. With respect to a monitoring plan, the current monitoring scheme and the information generated can be further utilized (Table 2).

**Table 2.** Assessment of the characteristics and design of the protected area (based on WCPA and WWF [40,41].

| Evaluation Element | Subject | Selected Criterion |
|---|---|---|
| Context | Legal status<br>*Does the protected area<br>have legal status?* | No legal protected area decree |
| | Area regulations<br>*Are unsuitable land uses and activities controlled?* | Although there are mechanisms to control land uses and activities, there are limitations to their effective implementation |
| | Application of regulations<br>*Are the regulations being applied satisfactorily?* | Staff have acceptable capacity to implement legislation and regulations. |
| | Demarcation of territorial boundaries<br>*Is the location of the boundaries known, and were they delineated in the field?* | Authorities know the area boundaries, but local people just know the approximate limits of the BL territory |
| | Natural resources inventory<br>*Is there sufficient information for the management of the area?* | Available information on critical habitats, species, and cultural values are insufficient to support planning and decision-making processes |
| Planning | Objectives of the area<br>*Are there established objectives?* | The area is being managed to achieve the objectives |
| | Design and extent of the area<br>*Is there a need to increase the area or implement biological corridors to achieve the objectives?* | The design of the area is suitable for the achievement of the primary objectives. |
| | Management plan<br>*Is there a management plan? Is it being implemented?* | There is a management plan |
| | Annual operative plan<br>*Is there an annual work plan?* | There is a working plan and activities are monitored against set targets. A high proportion of them are fulfilled |
| | Monitoring and evaluation<br>*Is there a research and monitoring program oriented towards the management of the area?* | There is an agreed monitoring and evaluation system in place, but the results are not systematically used for management. |
| Management | Biological importance: species | The area has few rare, threatened, or endangered species |
| | Biological importance: critical function habitat. | The protected area provides a habitat with a medium critical function |

*3.2. Infrastructure and Participation in the Protected Area*

Inputs: Although in the study area there are adequately trained personnel for the tasks entrusted to them, more personnel are needed, mainly in the accompaniment of the projects and local actors according to their profiles. The budget is sufficient to maintain the required operating capacity. The financing of the activities is supplemented by external financing from public and private, national, and foreign organizations (Table 3).

Processes: Related to the local populations and BL, there is communication, although it is not the strongest. Activities are carried out related to tourism, so they are followed up with local providers. There is a need to improve conditions for the tourism sector (Table 3).

Products: Infrastructure and services are appropriate to the current visitation levels but can be improved. It is necessary to have better defined and signposted routes, and to order some massive activities (Table 3).

The area comprising the BL provides a medium critical function habitat and is comprised of ecosystems that are less affected. The current budget is sufficient to meet all management needs and there is adequate and effective equipment and infrastructure for the current levels of visitation but there is room for further improvement. No fee is required to enter and there is a significant economic benefit to the local communities.

**Table 3.** Infrastructure assessment and participation based on WCPA & WWF [40,41].

| Evaluation Element | Subject | Selected Criterion |
|---|---|---|
| **Inputs** | Staff<br>*Is there sufficient staff to manage the protected area?* | The quantity of staff is insufficient for critical management activities |
| | Training<br>*Is there sufficient training for the staff?* | Staff training and skills are adequate for current challenges and future management |
| | Annual budget<br>*Is the current budget sufficient to manage the area?* | The current budget is sufficient to meet management needs |
| **Processes** | Education program<br>*Is there a planned program of education?* | There is a planned education and awareness program, but there are uncovered thematic and territorial areas |
| | Protected area and neighbors' relationships<br>*Is there cooperation with the neighbors of the protected area?* | Communication and cooperation between BL staff and the nearby property owners is desirable and positive |
| | Local communities<br>*Do the local communities (internal and external to the area) have access to decision making?* | The local communities participate directly in decision-making on the management of the protected area |
| | Tourism operators<br>*Do tourism operators contribute to the management of the protected area?* | There is excellent cooperation between staff and the tourism sector to improve the visitor experience, protect the cultural and natural values of the area, and resolve conflicts |
| **Products** | Infrastructure for visitors<br>***Is the infrastructure for visitors (tourists, pilgrims, etc.) sufficient?*** | Infrastructure and services are appropriate to the current visitation levels but can be improved |

### 3.3. Biocultural Landscape Model Stakeholder Interviews

Stakeholders mentioned that while most people know what a protected area is, the opposite is true for the BL area. In both cases there are perceptions that may differ from the correct ones. Even so, it is common knowledge that the protection of natural resources is important for the community. Among other relevant points, local stakeholders agree that there is little community participation and that decreeing certain territories as natural protected areas can generate conflicts or negative or positive consequences. It is stated that limitations in the use of natural resources, power scales, and individual interests are aspects that influence local heritage protection issues (Table 4).

**Table 4.** Type of response by theme related to BL.

| Theme/Question | Agree | Disagree | Both |
|---|---|---|---|
| 1—Knowledge about what a natural protected area is | 58% | 42% | |
| 2—Knowledge of what the BL model is | 42% | 58% | |
| 3—Consider the protection of natural resources important for the community | 100% | | |
| 4—There is government influence in the BL (positive or negative) | 67% | 25% | 8% |
| 5—The local participation in the BL planning processes is scarce | 58% | 34% | 8% |
| 6—There are negative consequences of natural resource protection decrees | 50% | 26% | 24% |
| 7—Economic alternatives are scarce in places where natural resources are protected by decree | 26% | 74% | |
| 8—Natural resources, ecosystems, and biodiversity conservation generates conflicts between actors | 66% | 26% | 8% |
| 9— Political actors make decisions without the participation of the local community | 66% | 34% | |
| 10—The scale of powers among local actors generates integration problems in the BL model | 58% | 42% | |
| 11—The different organizations collaborate without conflicts | 74% | 26% | |
| 12—Differences in the formation and opinions of key actors affect the interaction in the BL model | 50% | 42% | 8% |
| 13—Knowledge that natural protected areas can be accepted or rejected | 58% | 16% | 26% |
| 14—The nomination of a protected area causes use restrictions | 66% | 34% | |

### 3.4. Local Conflicts

There were six topics represented as codes related to conflicts. The four most frequent were: conflicts due to scale power, between stakeholders, natural protected area rejection, and negative effects resulting from a decree (Figure 2).

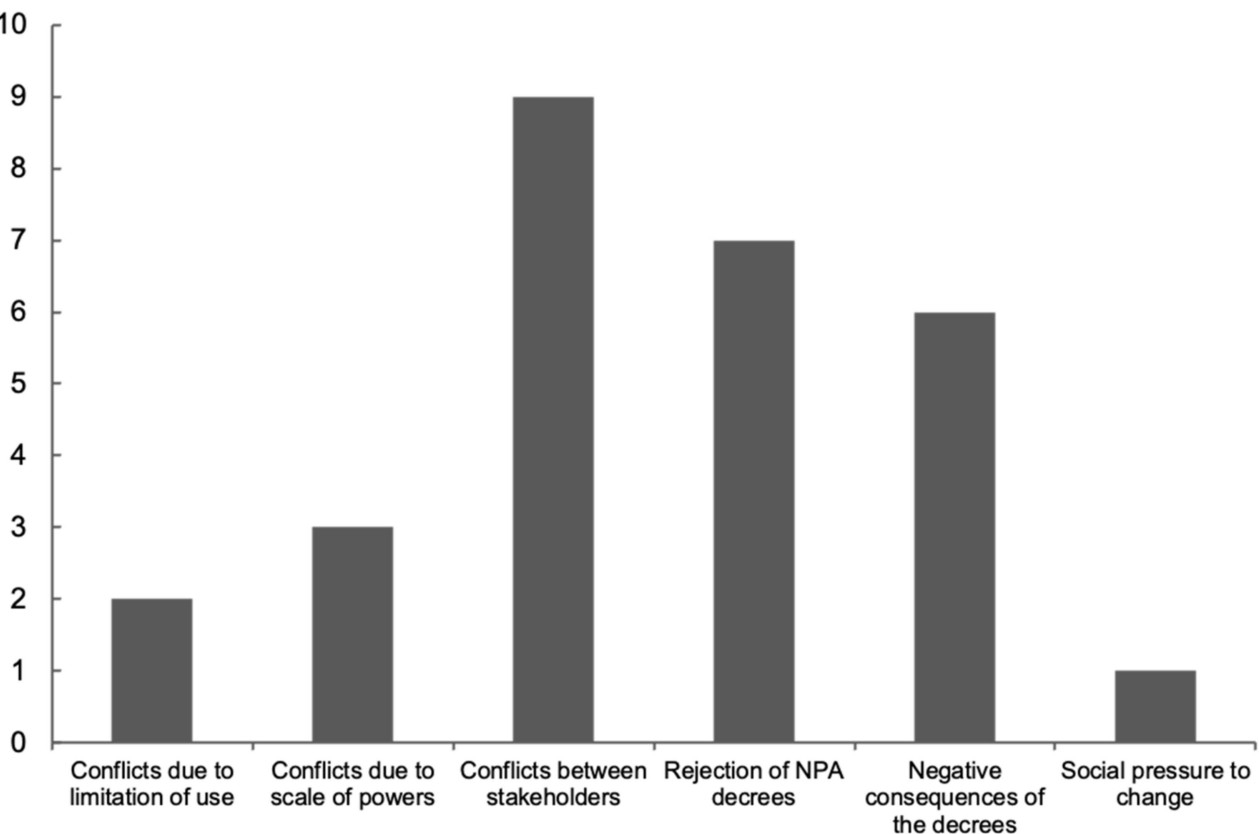

**Figure 2.** Frequency of responses by local conflict codes.

These conflicts are described next:

(a)  Conflicts between actors. This includes conflicts between those who prefer to continue using natural resources "as per usual" and those who prefer or are interested in conservation. The former argue that resources are necessary for survival and that differences in thinking are normal, as is the case in any social issue. The latter believe that a change is necessary.

(b)  Rejection of natural protected area decrees. There is little information on the subject, there are economic interests other than protection issues, the idea of what natural protected areas is wrong, and if natural resources are protected, local populations could be affected.

(c)  Conflicts due to power scales. There are three negative consequences detected: people are reluctant to change their way of thinking, they do not see benefits from conservation, and there are problems of land ownership

(d)  Negative consequences due to protection decrees. This group did not consider that there were transcendental conflicts, it was only mentioned that they could arise due to economic and knowledge differences, as well as the power and influence of large companies in the region.

*3.5. Conservation and Knowledge in Protected Areas*

The responses about Conservation and knowledge were distributed in 11 groups (Figure 3).

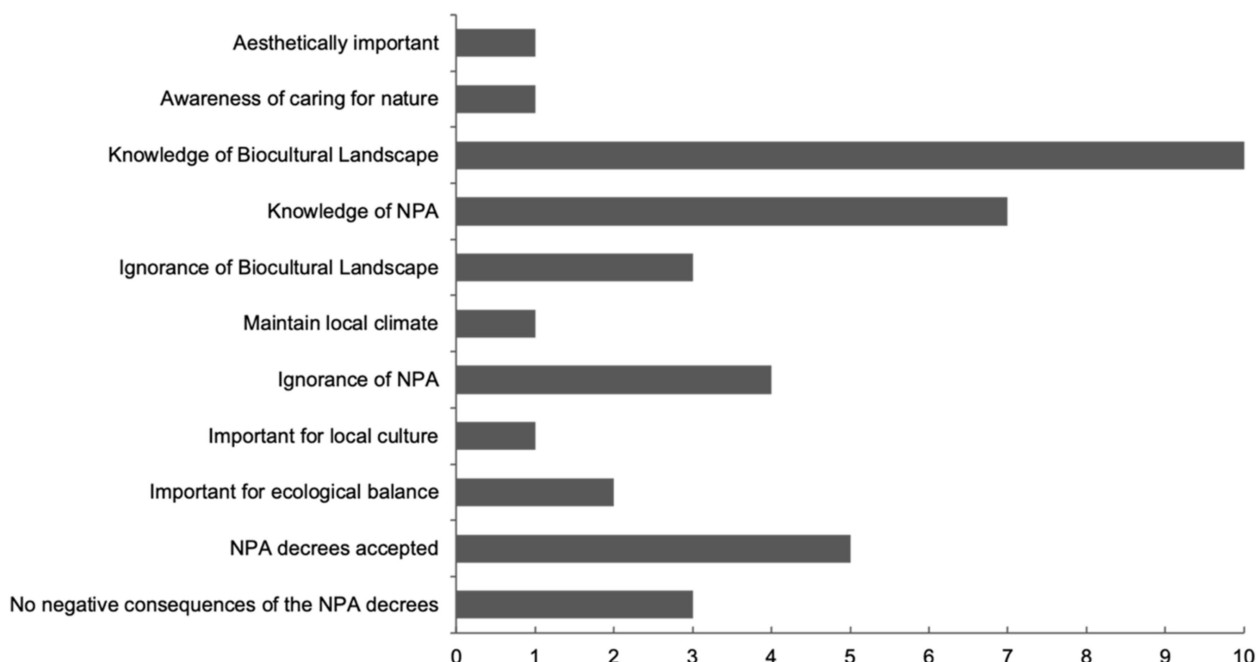

**Figure 3.** Frequency of responses by conservation and knowledge codes in protected areas (NPA).

The six most relevant codes in this group were selected for this theme and are next described:

(a) Knowledge of the concept of BL: The opinions state that the term is well known in the four municipalities involved, where those who know best about it are the producers.

(b) Knowledge of protected natural areas: It is perceived that there is this knowledge through the local working groups developed by the BL staff; however, it is not enough because they do not know the function of these areas or they only have partial ideas.

(c) Acceptance of the protection decrees: It is accepted since responsible use is encouraged, awareness has been raised and agreements created, and, also, more people have accepted it since they have observed that those who agree have obtained benefits.

(d) Lack of knowledge about natural protected areas: They mention that they have only heard about the topic without going deeper into it.

(e) There are no negative consequences: They consider that it is favorable for the environment since conservation is positive and local participation has been more active.

(f) Ignorance of the BL: This is because erroneous information has been distributed.

In general, the interviewees consider that those people who accept the decrees are aware of their benefits since they have encouraged a respectful use of nature, agreements have been reached and a collective conscience has been generated. Although most people in the study area know about the natural protected areas, misinformation is a determining factor, as they have misconceptions or partial ideas, and are unaware of their function and objective. For those who know about the BL area, it is due to their participation in the project (mainly producers, merchants, and local leaders).

*3.6. Territorial Management*

The nine most popular codes of a total of 16 (Figure 4) in the territorial management group, which proved to be the most extensive, were selected and are described below.

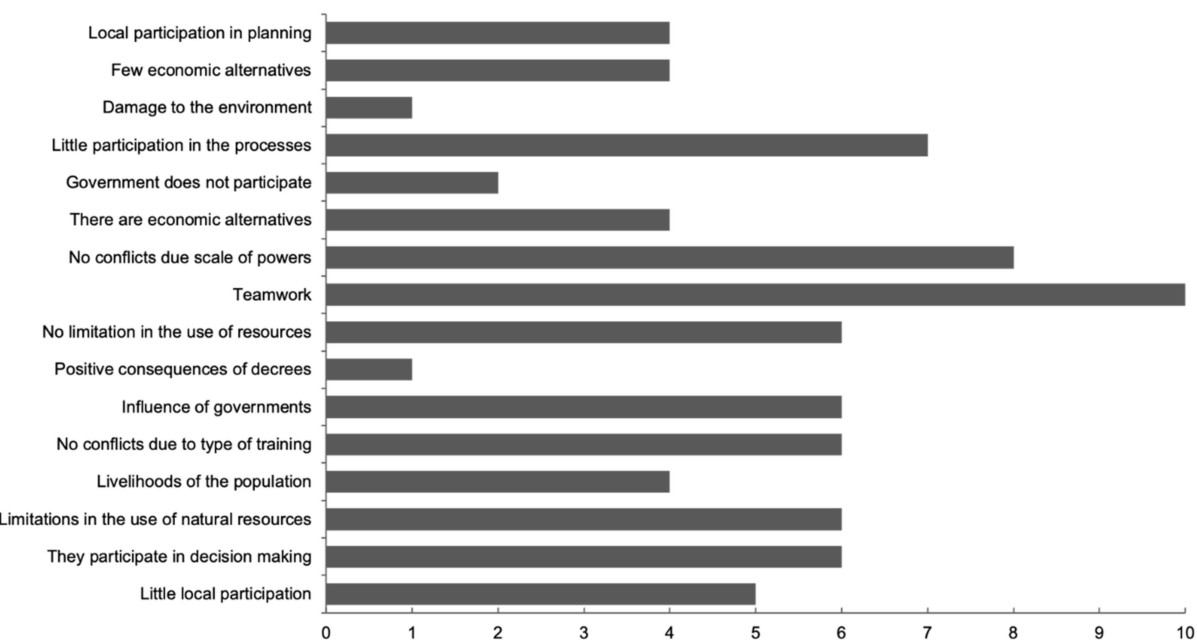

**Figure 4.** Frequency of responses by territorial management codes.

(a)   Teamwork: The participants in the BL model are willing to do so, so it is possible to work without conflict.

(b)   There are no conflicts over power scales: These are people with a group conscience in which specialists are integrated and contribute their knowledge so that there is equal participation without distinction.

(c)   Participation in the planning processes: It has been achieved, but it depends on the subject, since not all of them can be agreed upon, in addition, sometimes only the participation of trained personnel is possible.

(d)   Influence of governments: It exists positively since we have worked with different secretariats, directorates, and agencies, in addition to the existence of programs and projects that are added.

(e)   There are no conflicts due to types of training: All people, from their area of knowledge, contribute their ideas, so there is respect, group awareness, and equitable participation, which has also been possible thanks to the participation of specialists.

(f)   Limitation in the use of natural resources: Norms have been established for the use of resources, as well as ejido programs, which has made it possible to regulate some activities.

The interviewees commented that although there is a certain amount of rivalry in terms of power and the type of training of those involved in conservation issues, they have managed to control the situation because most of the activities are voluntary, where participation is diverse and everyone contributes their knowledge, different disciplines are integrated, and there is teamwork, both with the localities and the organizations. Limitations on the use of natural resources are due to norms and agreements that allow for their regulation. Regarding planning issues and local participation, it depends on the aspect being discussed, since some points are very specialized and it is difficult to reach a consensus on the decision, so only experts in the field participate. The communities depend on local resources, so they need to know the decrees, because of the need for them for their economic development and social dynamics, and this is an aspect that has been strengthened, as they have been able to learn more about the measures and regulations related to the project, all of which has awakened environmental awareness in the region.

## 4. Discussion and Conclusions

Local communities are gradually becoming more involved in the BL area's planning processes, although rural populations usually have limited knowledge about the objectives of a natural protected area and its functioning; they perceive its importance but do not identify direct benefits [44]. Latin American protected areas have reported that many of the erroneous ideas about them are frequently assumed as real, creating conflicts and affecting the collaborative work [11]. This causes a different perception of the BL model affecting local participation and can complicate the acceptance of participatory conservation models. In this sense, it is necessary to perform effective interventions to improve the planning and communicating process. This would pose a challenge, since, more than a benefit, people may consider it a threat that natural areas, even cultural ones, may be protected, since their social dynamics are developed based on these, commercially, economically and traditionally.

In addition to the misconception of the model, conflicts arise due to the diversity of interests that certain local stakeholders have, so progress could be slow or simply non-existent. In theory, everyone can participate in the BL model, but in specific cases only the members of relevant groups can participate since specialized intervention is required, a fact that could be interpreted as a simulation [21]. As Carvalho proposes [45], to be successful in the conservation and management of protected areas, it is necessary to ensure that people participate actively, not simply integrating their knowledge and expertise when it is opportune.

A virtuous cycle is developing in the BL model, the formal and informal agreements to protect the biocultural landscape have incentivized local participation, enhancing the information and management of the area as observed by other authors [14,15].

The main challenge of the BL model continues to be social participation, although it has been improved, if vertical implementation of their strategies does not effectively consider community participation [28].

Something that is undeniable is that people have acquired greater environmental awareness, since those who actively participate see the protection decrees as a positive measure, because they do not find a limitation in the use of nature, which evidently reflects an important advance in conservation. This differs from what usually happens with traditional natural protected areas [11].

The BL model has made clear progress by monitoring its activities, knowing the flora and fauna of the region and the commercial activities with greatest potential. However, there are aspects that need to be worked in greater depth: clarifying what the natural areas consist of, clarifying what their rules and regulations are, and communicating adequately will all allow local people to become more involved in conservation issues. Increasing participation can also contribute to a greater dissemination of knowledge that would reinforce environmental awareness.

Adaptive governance leads to flexible collaborations, which focus on learning and decision-making processes that involve multilevel actors, with the objective of negotiating and coordinating issues such as landscape and seascape management [15]. When actors with different interests coexist, adaptive governance enables understanding and improving governance responses to challenges faced in sustainability issues, including learning and collaboration with sectors and scales that have a shared vision through monitoring and information exchanges, networking, and conflict resolution.

Related to the questions that guided the research, we can answer that stakeholders' interaction and collaboration evidence that the BL model surpasses the traditional model of protected natural areas implemented by governments, but there are still challenges related to local participation. Although to some extent the BL model repeats the linear and hierarchical approach for conservation, since it was designed at the beginning by governments of different scales together with non-governmental organizations, and it was not prompted by community, it was not imposed. It has acquired solidity through collaborative work as a form of participation of different key actors that guide the actions.

We can also answer that BL area functions in an innovative governance context where the social participation and collaboration are distinctive, but it has been a slow process.

Addressing issues in the interest of all stakeholders, as proposed by Zhang [46], can help in avoiding conflicts and in improving the local development and the livelihood of communities. The theory said years ago that conservation should be reached if participative forms of conservation were performed; we underline that conservation must be done based upon biocultural approaches in a real participative exercise.

As to whether the model has achieved the BL conservation objectives in this area, it is too soon to see results on particular species, ecosystems, or social groups, but, as stated at the beginning, social participation is required for conservation [9], so it is on its way. Future research should study these particular indicators to evidence positive field changes in biocultural conservation.

Finally, one of the limitations of this research that we recognize is that the World Bank/WWF Management Effectiveness Tracking Tool [41] adapted here to evaluate the BL area in Jalisco was intended to facilitate reporting on progress in management effectiveness from the BL staff perspective. The stakeholders included in the study were pro-BL due the tool used [42], so in the process we may not have considered key stakeholders who did not agree with the model or who are affected by the achievement of its objectives as proposed by Benn et al. [43]. To avoid bias, it should not be the only analysis conducted for adaptive management purposes

**Author Contributions:** Conceptualization, R.M.C.-D.; Data curation, O.A.M.I.; Formal analysis, O.A.M.I.; Funding acquisition, O.A.M.I.; Investigation, O.A.M.I., R.M.C.-D. and M.L.B.-O.; Methodology, O.A.M.I., R.M.C.-D. and R.T.A.-S.; Project administration, R.M.C.-D.; Resources, R.T.A.-S.; Software, O.A.M.I.; Supervision, R.M.C.-D. and M.L.B.-O.; Validation, O.A.M.I. and M.L.B.-O.; Visualization, M.L.B.-O. and R.T.A.-S.; Writing—original draft, O.A.M.I. and R.M.C.-D.; Writing—review & editing, O.A.M.I., R.M.C.-D., M.L.B.-O. and R.T.A.-S. All authors have read and agreed to the published version of the manuscript.

**Funding:** We would like to thank the Rufford Foundation for partially funding this work through project #30148-1 "Biocultural Landscape as a model of conservation and community participation in the western Sierra de Jalisco".

**Institutional Review Board Statement:** Not applicable.

**Informed Consent Statement:** Informed consent was obtained from all subjects involved in the study.

**Data Availability Statement:** The data presented in this study are available on request from the first author or corresponding author.

**Conflicts of Interest:** The authors declare no conflict of interest. The funders had no role in the design of the study; in the collection, analyses, or interpretation of data; in the writing of the manuscript, or in the decision to publish the results.

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
