# Peer review of "Challenges for Social Participation in Conservation in the Biocultural Landscape Area in the Western Sierra of Jalisco"

_land, doi:10.3390/land11081169_

Round 1

Reviewer 1 Report

1) This paper appears to be an interesting contribution to research on ‘social participation in conservation in Biocultural Landscape`;

2) Its academic value makes it suitable for the international and cross-disciplinary, peer-reviewed, open access journal Land (ISSN 2073-445X); 

3) However, there are some spell checking and writing issues that must be addressed by the authors. See, for instance, the “examples” listed below (underlined text):

  • Lines 65-67, CHECK TEXT, how many questions are we talking about: ‘where the general question??? arises: Does this model surpass the traditional model of protected natural areas implemented by governments? What challenges does it face related to social participation?’;
  • Lines 176-180, CHECK (repetitive??) TEXT: ‘The first contact was through the staff responsible for BL management and with a chain sampling. The first contact was through the staff responsible for BL management and with a chain sampling??? including local and regional actors related to the BL area of the four municipalities, respecting the diversity of roles in the area (Table 2).’;
  • Table 3, CHECK TYPOS: ‘Assesment??? of the characteristics and design of the protected área???’;
  • Table 3, CHECK TEXT: ‘The boundaries of the area are not??? but they are known by the management authority. Local people??? Local people know the approximate limits of the BL territory’;
  • Lines 284-285, CHECK ENGLISH: ‘but those??? who know the most about it is??? the producers.’;
  • Line 304, CHECK TEXT, a ‘period’ appears to be missing: ‘people interested in the project, mainly producers, merchants, and local leaders???’;
  • Lines 306-307 and 312-326, CHECK TEXT: ‘In the territorial management map, which proved to be the most extensive, the nine??? most popular of a total of 16 codes (Figure 4) were selected and are described below’.

Reviewer 2 Report

The manuscript needs a profound review. Some aspects are not sufficiently considered. 

The hierarchical approach implies poor local participation. But for who?

Could you define/mention who the stakeholders are?. They are not clearly specified. 

Table 1 must be redefined, showing the total population and main economic activities of the area, and non by municipalities.

Also, the authors must note the concrete natural and cultural heritage. The description of these will help readers to understand in a good way the paper.

Tables 3 and 4 are excessively long; can these be reduced? What about the source of these?. Moreover additionally, the explanations of these tables and graphics must be extended. Less schematic and more extended explanations of graphics and tables. The extreme case is Figure 3. 

State of the art must be extended and explicitly mentioned. What about other experiences of social participation in the conservation of biocultural landscapes in other continents?: i.e. in Europe (Sustainable Tourism European Letter or LEADER approach, i.e.).  

And finally, the Discussion paragraph is not such a Discussion because it lacks references and comparisons with similar studies.

Some minor corrections related to the repetition of words:

- Page 1: "about the concept of biocultural concept"

- Has the model achieved social participation to achieve ...

- Page 6: "local people local people"

- Include the North symbol on the map. 

Reviewer 3 Report

This manuscript deals with relevant issues, and it presents scientific soundness.

However, I recommend mainly the following improvements:

  1. In the abstract, I miss a deeper description of the methods and synthetic conclusions.
  2. I don’t see the classical structure: introduction, literature review, …
  3. The I suggest comparing your study findings with the earlier studies in global context wherever possible.
  4. Discussion needs to be improved and supported well by literature. In the last part of the discussion section, I would like to see the main limitations of the study (1 or 2 sentences).
  5. Add directions for future studies at the end of conclusion section.
  6. The conclusions are trivial and do not add much to the theory and practice of

Good luck!

Round 2

Reviewer 2 Report

The authors revised all the comments and improved the manuscript substantially. Anyway, some relevant points must be explained before the publication of the text:

Firstly, and mostly, the contrariety between the authors hypothesized, “traditional natural protected areas generally designed by governments linearly and hierarchically excluding the local participation”, and the selected stakeholders, mostly those of “local governments”: “municipal government official…”. Authors must clarify/explain in the text this explicit contradiction. 

Km2 and/or hectares (line 165)

Lines 172-179 are excessively generic; these must be explained in more detail, and adding scientific names of the species of flora and fauna. 

Could the authors add a sentence below tables 3 and 4? noting: Source: The authors. Or: Source: Own elaboration. 

Reviewer 3 Report

An improvement indeed!